# Bulk Viscosity of Dilute Gases and Their Mixtures

**Bhanuday Sharma** [1], **Rakesh Kumar** [1,*] and **Savitha Pareek** [2]

1   Department of Aerospace Engineering, Institute of Technology Kanpur, Kanpur 208016, Uttar Pradesh, India; bhanuday@iitk.ac.in
2   DELL HPC and AI Innovation Lab, Bengaluru 560093, Karnataka, India; savitha.pareek@dell.com
*   Correspondence: rkm@iitk.ac.in; Tel.: +91-9411396999 or +91-5122596301

**Abstract:** In this work, we use the Green–Kubo method to study the bulk viscosity of various dilute gases and their mixtures. First, we study the effects of the atomic mass on the bulk viscosity of dilute diatomic gas by estimating the bulk viscosity of four different isotopes of nitrogen gas. We then study the effects of addition of noble gas on the bulk viscosity of dilute nitrogen gas. We consider mixtures of nitrogen with three noble gases, viz., neon, argon, and krypton at eight different compositions between pure nitrogen to pure noble gas. It is followed by an estimation of bulk viscosity of pure oxygen and mixtures of nitrogen and oxygen for various compositions. In this case, three different composition are considered, viz., 25% $N_2$ + 75% $O_2$, 50% $N_2$ + 50% $O_2$, and 78% $N_2$ + 22% $O_2$. The last composition is aimed to represent the dry air. A brief review of works that study the effects of incorporation of bulk viscosity in analysis of various flow situations has also been provided.

**Keywords:** bulk viscosity; molecular dynamics; Green–Kubo relations

## 1. Introduction

Two coefficients of viscosity, viz., shear ($\mu$) and bulk viscosity ($\mu_b$), are required to describe the flow of an isotropic, Newtonian, and homogeneous fluid in a continuum framework. In commonly encountered flow problems, it is sufficient to account for only the effects of shear viscosity and ignore the bulk viscosity. Therefore, bulk viscosity is relatively less studied as compared to shear viscosity. In fact, for several years, the existence of bulk viscosity had been questioned [1]. However, numerous studies [2–29] in the past three decades have been carried out on estimation of bulk viscosity, and it is now well established that fluids possess nonzero bulk viscosity. Several studies [30–42] have also been conducted to study the effects of accounting bulk viscosity in these flow situations, and these suggest that this transport property may play an important role in capturing physics of several flow situations. However, the scope and accuracy of these studies are limited by available values of the bulk viscosity of fluids. In contrast to shear viscosity, precise and accurate values of bulk viscosity are rarely available and typically a wide variation in the estimates of the latter can be observed in the literature.

In this work, we first briefly review the works that focus on mechanisms responsible for bulk viscosity effects, and then summarize a few studies on effects of bulk viscosity in various flow problems. Then, we use the Green–Kubo method in equilibrium molecular dynamics framework to calculate the bulk viscosity of dilute gases and their mixtures. We first study the effects of atomic mass on the bulk viscosity of dilute gas. To this end, we have estimated the bulk viscosity of three isotopes of nitrogen. It is followed by a study on the effects of addition of noble gases to dilute nitrogen by calculating the bulk viscosity of mixtures of dilute nitrogen gas and noble gases. Furthermore, we have estimated the bulk viscosity of pure oxygen and the mixture of nitrogen and oxygen (including a particular case of dry air).

The rest of the paper is organized as follows. A brief overview of mechanisms of bulk viscosity from the perspective of both dilute and dense fluids is given in Section 2. Section 3

reviews past works on importance of bulk viscosity effects in various flow situations. The details of numerical method and molecular model are given in Section 4. The results and corresponding discussion is provided in Section 5. Concluding remarks are made in Section 6.

## 2. Mechanism of Bulk Viscosity

Bulk viscosity is a macroscopic manifestation of the microscopic relaxation phenomenon. On the basis of the type of energy involved in relaxation, it can be classified into two components. Nettleton [43] named these two components as apparent and intrinsic bulk viscosity. Apparent bulk viscosity is related to the finite rate of energy exchange among translational and internal degrees of freedom. Therefore, this is the primary mechanism of bulk viscosity in polyatomic fluids, e.g., $N_2$, and $CO_2$. On the other hand, intrinsic bulk viscosity is related to the relaxation of potential energy. This mechanism is the primary source of bulk viscosity in dense gases and liquids.

### 2.1. Apparent Bulk Viscosity

This mechanism was first suggested by Herzfeld and Rice [24]. Whenever a gas is compressed or expanded, it gains or loses energy by means of work. However, this exchange of energy only affects the translational mode of motion. That means the rise or fall of translational energy is instantaneous. On the other hand, the internal modes, i.e., rotational and vibrational modes, receive/transfers energy from/to translational mode by means of intermolecular collisions. Hence, the energy of internal modes does not alter instantaneously but at a finite rate. The mechanical pressure is only caused by translational motion, whereas the thermodynamic pressure is the mechanical pressure of the system when brought to equilibrium through an adiabatic process [44]. Since, during change of volume, the instantaneous translational energy of the system is not same as that if the system is adiabatically brought to equilibrium, the mechanical and thermodynamic pressures differ. This difference between mechanical and thermodynamic pressure causes nonzero bulk viscosity.

### 2.2. Intrinsic Bulk Viscosity

This mechanism was first proposed by Hall [45] in 1948. Consider a compression of dense fluid such as liquid water. During compression, two different processes take place. The first one is that molecules are brought uniformly closer together. It can be called molecular compression, and it is an almost instantaneous process. The second process is that molecules are rearranged or repacked more closely. Hall identified this process as configurational or structural compression. This process involves the breaking of intermolecular bonds (e.g., hydrogen bonds) [46] or flow past energy barriers, which stabilizes the equilibrium configuration. This is a finite rate process. Thus it is of relaxational nature and is a source of nonequilibrium. This mechanism of bulk viscosity is present in all fluids including monatomic gases. Hence, monatomic gases at atmospheric conditions have a small ($\mathcal{O}(10^{-10})$ Pa s) but non-zero bulk viscosity. It should also be noted that at hypothetical dilute gas conditions, the bulk viscosity of monatomic gases is considered to be absolute zero.

## 3. Applications of Bulk Viscosity

Bulk viscosity effects become important when either the $\nabla \cdot \vec{u}$ is high (e.g., inside a shock wave), or when fluid is compressed and expanded in repeated cycles such that the cumulative effect of the small contributions from each cycle is no more negligible (e.g., sound wave) [47], or when the atmosphere consists of the majority of those gases, such as $CO_2$, which exhibit a large bulk viscosity [48], or when results of interest might get affected by even small disturbances, e.g., the study of Rayleigh–Taylor instability [49]. In such cases, it becomes necessary to account for the bulk viscosity terms in the Navier–Stokes equation.

Several researchers have investigated the effects of the incorporation of bulk viscosity in analytical or CFD studies of various flow scenarios. Emanuel et al. [48,50–52] reviewed bulk viscosity and suggested that the effects of bulk viscosity should be accounted for in the study of high-speed entry into planetary atmospheres. They observed that the inclusion of bulk viscosity could significantly increase heat transfer in the hypersonic boundary layer [48]. Chikitkin [53] studied the effects of bulk viscosity in flow past a spacecraft. They reported that the consideration of bulk viscosity improved the agreement of velocity profile and shock wave thickness with experiments. Shevlev [54] studied the effects of bulk viscosity on $CO_2$ hypersonic flow around blunt bodies. The conclusions of their study were in line with that of Emanuel. They suggested that incorporation of bulk viscosity may improve predictions of surface heat transfer and other flow properties in shock layer.

Elizarova et al. [55] and Claycomb et al. [56] carried out CFD simulations of normal shock. They found that including bulk viscosity improved the agreement with experimental observations for shock wave thickness. A recent study by Kosuge and Aoki [30] on shock–wave structure for polyatomic gases also confirms the same. Bahmani et al. [57] studied the effects of large bulk to shear viscosity ratio on shock boundary layer interaction. They found that a sufficiently high bulk to shear viscosity ratio can suppress the shock-induced flow separation. Singh and Myong [31] studied the effects of bulk viscosity in shock–vortex interaction in monatomic and diatomic gases. They reported a substantially strengthened enstrophy evolution in the case of diatomic gas flow. Singh et al. [32] investigated the impact of bulk viscosity on the flow morphology of a shock-accelerated cylindrical light bubble in diatomic and polyatomic gases. They found that the diatomic and polyatomic gases have significantly different flow morphology than monatomic gases. They produce larger rolled-up vortex chains, various inward jet formations, and large mixing zones with strong, large-scale expansion. Touber [35] studied the effects of bulk viscosity in the dissipation of energy in turbulent flows. He found that large bulk-to-shear viscosity ratios may enhance transfers to small-scale solenoidal kinetic energy and, therefore, faster dissipation rates. Riabov [58] questioned the ability of bulk viscosity to model spherically expanding nitrogen flows in temperature range 10 to 1000 K by comparing results to Navier–Stokes equations to relaxation equation. He reported that the bulk viscosity approach predicts much thinner spherical shock wave areas than those predicted by relaxation equations. Moreover, the distributions of rotational temperature along the radial direction predicted by the bulk viscosity approach had neither any physical meaning nor matches with any known experimental data for expanding nitrogen flows.

Fru et al. [59] performed direct numerical simulations (DNS) study of high turbulence combustion of premixed methane gas. They found that the incorporation of bulk viscosity does not impact flame structures in both laminar and turbulent flow regimes. Later, the same group extended their study to other fuels, viz., hydrogen, and synthetic gas. In this study [60], they found that though flame structures of methane remained unchanged before and after incorporation of bulk viscosity, the same for hydrogen and syngas showed noticeable modifications. Sengupta et al. [49] studied the role of bulk viscosity on Rayleigh Taylor instability. They found that the growth of the mixing layer depends upon bulk viscosity. Pan et al. [33] has shown that bulk viscosity effects cannot be neglected for turbulent flows of fluids with high bulk to shear viscosity ratio. They found that bulk viscosity increases the decay rate of turbulent kinetic energy. Boukharfane et al. [36] studied the mechanism through which bulk viscosity affects the turbulent flow. They found that the local and instantaneous structure of the mixing layer may vary significantly if bulk viscosity effects are taken into account. They identified that the mean statistical quantities, e.g., the vorticity thickness growth rate, do not get affected by bulk viscosity. On this basis of their study, they concluded that results of refined large-eddy simulations (LES) might show dependence on the presence/absence of bulk viscosity, but Reynolds-averaged Navier–Stokes (RANS) simulations might not, as they are based on statistical averages.

Connor [34] studied the effects of bulk viscosity in the compressible turbulent one-, two-, and three-dimensional Couette flows through DNS simulations. The objective of the

study was to test whether invoking the Stokes' hypothesis introduces significant errors in the analysis of compressible flow of solar thermal power plants, and carbon capture and storage (CCS) compressors. They found that most of the energy is contained in the solenoidal velocity for both CCS and concentrated solar power plants. Therefore, assuming bulk viscosity to be zero does not produce any significant errors, despite the compressors operating at supersonic conditions. However, bulk viscosity effects may become significantly close to the thermodynamic critical point.

Billet et al. [61] showed that the inclusion of bulk viscosity in CFD simulations of supersonic combustion modifies the vorticity of the flow. Lin et al. [38,39] have shown that acoustic wave attenuation in CFD simulations can be accounted for by incorporating bulk viscosity. Nazari [37] studied the influence of liquid bulk viscosity on the dynamics of a single cavitation bubble. They reported that bulk viscosity significantly affects the collapse phase of the bubble at high ultrasonic amplitudes and high viscosities. High bulk viscosity values also altered the maximum pressure value inside the bubble.

In all of these applications and beyond, precise values of bulk viscosity are needed. However, in contrast to shear viscosity, which is a well studied subject, neither the nature of the bulk viscosity is well understood, nor widely accepted values of bulk viscosity are available even for common gases, such as nitrogen, oxygen, and air. To this end, the present work aims to fill some of this research gap by calculating bulk viscosity of dilute gases and their mixtures through a molecular dynamics approach.

## 4. Molecular Model and Simulation Details

In the present work, we have performed molecular dynamics (MD) simulations and applied the Green–Kubo formulation to calculate viscosity coefficients. A brief review of the Green–Kubo method and corresponding simulation details are discussed in following paragraphs.

### 4.1. Green–Kubo Method

There have been several methods proposed in the literature for the estimation of bulk viscosity. A brief survey of these methods can be found in Refs. [11,12]. In our previous publications [11,12], we have systematically studied two simulation methods to estimate bulk viscosity, viz., nonequilibrium molecular dynamics (NEMD) based continuous expansion/compression method and equilibrium molecular dynamics (EMD)-based Green–Kubo method. It was established that both the methods predict accurate bulk viscosity values; however, the Green–Kubo method requires less computational resources [12]. Therefore, the same has been used in this present study to investigate the nature and quantitative value of the bulk viscosity of dilute gases and their mixtures. This method is based on Green–Kubo relations [Equations (1) and (2)], which are derived from fluctuation-dissipation theorem and express viscosity coefficients in terms of the integral of auto-correlation functions of components of pressure tensor.

$$\mu = \lim_{t \to \infty} \frac{\mathcal{V}}{k_B T} \int_0^t \langle P_{ij}(t') \, P_{ij}(0) \rangle dt' \tag{1}$$

$$\mu_b = \lim_{t \to \infty} \frac{\mathcal{V}}{k_B T} \int_0^t \langle \delta P(t') \, \delta P(0) \rangle dt' \tag{2}$$

Here, $k_B$ is the Boltzmann constant, $\mathcal{V}$ is volume, $T$ is temperature, $P_{ij}(t')$ is the instantaneous value of $ij^{\text{th}}$ off-diagonal element of the pressure tensor at a time $t'$, and the angle bracket represents the ensemble average. In the Equation (2), the $P(t')$ is an instantaneous value of the average of three diagonal terms of pressure tensor at a time $t'$, i.e., $P(t') = \frac{1}{3}[P_{ii}(t') + P_{jj}(t') + P_{kk}(t')]$. The fluctuations, $\delta P(t')$, is deviation of mean pressure from equilibrium pressure, i.e., $\delta P(t') = P(t') - P_{eq}$; where $P_{eq}$ is equilibrium pressure of the system, and it is calculated by time average of $P(t')$ over a long time.

### 4.2. Molecular Model

In the present work, all the molecules are modeled using standard 12–6 Lennard–Jones potential, $V$, given as follows:

$$V(r_{ij}) = \begin{cases} 4\varepsilon\left[\left(\dfrac{\sigma}{r_{ij}}\right)^{12} - \left(\dfrac{\sigma}{r_{ij}}\right)^{6}\right], & \text{if } r_{ij} < r_{\text{cut-off}} \\ 0 & \text{if } r_{ij} > r_{\text{cut-off}} \end{cases} \tag{3}$$

where, $r_{ij}$ is the interatomic distance between atom $i$ and atom $j$, and $\sigma$ and $\epsilon$ are the Lennard–Jones parameters. In the present work, we have simulated nitrogen, neon, argon, krypton, and oxygen gases. The details of $\sigma$ and $\epsilon$ used for nitrogen, neon, argon, and krypton are given in Table 1. For oxygen, we have evaluated eight molecular potentials available in the literature and then used one of them for final calculations. The details of these potentials are given in Table 2. In order to model the interatomic interaction between unlike molecules, the Lorentz–Berthelot combination rule given by Equations (4) and (5) is used [4].

$$\sigma_{12} = \frac{1}{2}(\sigma_{11} + \sigma_{22}) \tag{4}$$

$$\epsilon_{12} = \sqrt{\epsilon_{11}\,\epsilon_{22}} \tag{5}$$

To perform MD simulations, a cubical simulation box with periodic boundaries is generated. In the initialization step, 200 molecules of the desired gas are randomly placed in the domain and given initial velocities sampled from the equilibrium Boltzmann distribution corresponding to the target temperature. After initialization, the system is equilibrated first in canonical and then in the microcanonical ensemble for $10^4$ and $10^6$ time steps, respectively. Finally, the production run is performed for $3 \times 10^8$ time steps in microcanonical ensemble. A time step size of 1 fs is used, and time integration is performed using the standard velocity-Verlet algorithm.

To calculate viscosity values accurately, the time-decomposition scheme by Zhang et al. [62] is used in the Green–Kubo framework. In this approach, the autocorrelation function required for the Green–Kubo method is calculated from the average of 120 trajectories, each of which is 300 ns in length. Each of these trajectories is initialized with macroscopically same but different microscopic state. These trajectories differed from each other in terms of the random number seed used for sampling of initial velocities from equilibrium Boltzmann distribution in the initialization step. Random number seeds used for sampling the initial velocities of the molecules are 777001, 777002, 777003, . . . , 777120. Rigid rotor approximation is applied everywhere except stated otherwise.

The open-source molecular dynamics simulation software LAMMPS [63] has been used for performing molecular dynamics simulations. The open-source plotting software VEUSZ [64] has been used for plotting graphs.

**Table 1.** Molecular models for nitrogen and noble gases.

| Gas | Source | $\epsilon/k_B$ [K] | $\sigma$ [Å] | Bond-Length [Å] |
|---|---|---|---|---|
| Nitrogen | Tokumasu et al. [65] | 47.202 | 3.17 | 1.098 |
| Neon | Vrabec et al. [4] | 33.921 | 2.8010 | – |
| Argon | Vrabec et al. [4] | 116.79 | 3.3952 | – |
| Krypton | Vrabec et al. [4] | 162.58 | 3.6274 | – |

**Table 2.** Molecular models for oxygen gas.

| Source | $\epsilon/k_B$ [K] | $\sigma$ [Å] | Bond-Length [Å] |
|---|---|---|---|
| CHARM [66] | 60.389 | 3.029 | 1.208 |
| CHARM-2 [66] | 60.389 | 3.029 | 1.12 |
| Javananien et al. [67] | 48.458 | 3.13 | 1.016 |
| Bouanich et al. [68] | 52.583 | 3.0058 | 1.21 |
| Cordeiro et al. [69] | 35.0 | 2.6 | 1.21 |
| Fishcher et al. [70] | 43.659 | 3.09 | 1.016 |
| Porrini et al. [71] | 52.331 | 2.96 | 1.12 |
| Victor et al. [72] | 67.0 | 3.57 | 1.214 |

## 5. Results and Discussion

### 5.1. Effect of Atomic Mass: Bulk Viscosity of Isotopes of $N_2$

Bulk viscosity in dilute nitrogen gas at temperatures less than 800 K is primarily due to the finite rate of energy exchange between translational and rotational modes. The ease of rotation of a molecule depends on the moment of inertia, and hence, on the mass of constitutive atoms and their distribution in the molecule. Therefore, it is expected that different isotopes would show different values of bulk viscosity, even though their interaction with neighboring molecules remains the same. To study this dependency of bulk viscosity on the mass of nuclei, we have calculated the bulk viscosity of four isotopes of nitrogen molecule. In the first three molecules, both the nitrogen atoms have the same atomic mass, viz., 13.0057, 14.0067, and 15.0001, for three molecules. In the fourth molecule, the two nitrogen atoms have different atomic masses, i.e., 13.0057 and 14.0067. For each of these molecules, both shear and bulk viscosities are calculated at 1 bar at four different temperatures. Figure 1a,b plot the variation of obtained viscosity values against temperature and compare them with the available experimental data for $_7^{14}$N from Refs. [15,19,20,73].

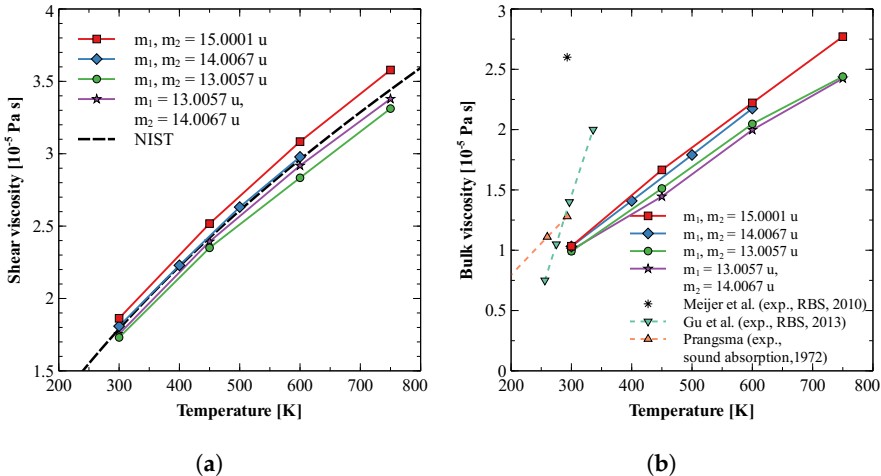

|  (a)  |  (b)  |

**Figure 1.** Bulk and shear viscosity of different isotopes of nitrogen gas. Here, $m_1$ and $m_2$ are masses of two atoms of the nitrogen molecule [15,19,20,73].

From Figure 1b, it can be observed that for the first group, where both the nuclei have the same atomic mass, bulk viscosity increases with an increase in moment of inertia (see Table 3). A possible explanation for this behavior could be that an increase in atomic mass increases the moment of inertia, making any change in rotational energy/velocity more difficult. Therefore, translational to rotational and rotational to translational energy exchange is slowed down. Furthermore, the bulk viscosity of isotope with heterogeneous nuclei is found to be smaller than the homogeneous ones. The probable reason is the fact

that heterogeneous mass distribution causes shifting of the center of mass, which makes rotation easier.

**Table 3.** Moment of inertia of molecules considered in Figure 1b.

| $m_1$ [u] | $m_2$ [u] | $I$ [u-Å$^2$] |
|---|---|---|
| 13.0057 | 13.0057 | 7.83986197 |
| 14.0067 | 14.0067 | 8.44326677 |
| 15.0001 | 15.0001 | 9.04209028 |
| 13.0057 | 14.0067 | 8.17510498 |

### 5.2. Bulk Viscosity of Mixture of Nitrogen with Noble Gas

In the process of understanding the effects of impurity (i.e., addition of another gas) on bulk viscosity of dilute gases, a simplest mixture would be a mixture of a diatomic and monatomic gas; as the dilute monatomic gases do not have rotational or vibrational degrees of freedom, their mixture would also show negligible bulk viscosity. In this work, we have estimated the bulk viscosity of mixtures of dilute nitrogen gas and three noble gases, i.e., neon, argon, and xenon. For each combination, bulk viscosity values were calculated at 300, 450, 600, and 750 K for thirteen different compositions, viz., 0, 5, 10, 20, 30, 40, 50, 60, 70, 80, 90, 95, and 100%. Figure 2 shows the obtained results. It can be observed that the bulk viscosity increases with temperature. It can also be observed that the small amount (<5%) of impurity of noble gas does not alter bulk viscosity value significantly. However, a small amount (5%) of nitrogen in 95% noble gas does make a noticeable increase in the bulk viscosity of mixture.

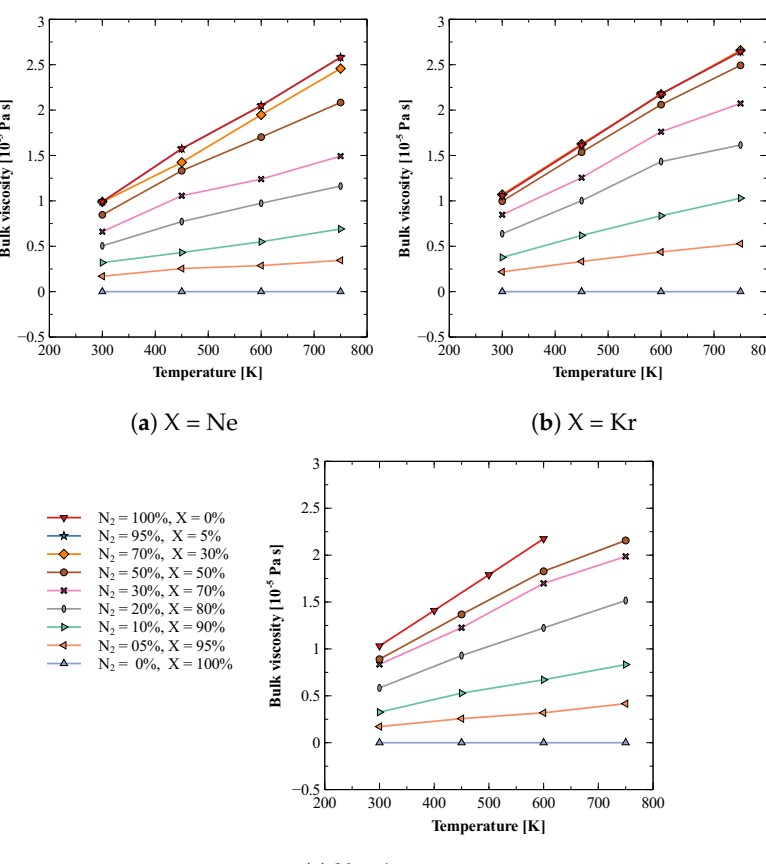

**Figure 2.** Bulk viscosity of different compositions of mixture of nitrogen ($N_2$) and noble gases (X). (**a**) $N_2$ + Ne (**b**) $N_2$ + Kr (**c**) $N_2$ + Ar.

### 5.3. Bulk Viscosity of $O_2$

Oxygen is the second-largest constituent (21%) of earth atmosphere after nitrogen (78%). Therefore, it becomes important to understand the nature of bulk viscosity of this gas before we study the bulk viscosity of mixture of oxygen and nitrogen. There are several atom–atom models available for modeling oxygen molecule in molecular dynamics simulations. Here, we have considered eight models and calculated their shear and bulk viscosities. The details of these models, e.g., $\sigma$ and $\epsilon$, are given in Table 2. Figures 3a,b show the results obtained for shear and bulk viscosity respectively and their comparison with the available experimental data of Refs. [73–77]. A close inspection of these figures shows that none of the models reproduces both shear and bulk viscosity values agreeing with experimental data simultaneously. The models by Bouanich et al. [68] and Fischer et al. [70] gives shear viscosity in close agreement to that of NIST [73] data but fails to reproduce bulk viscosity values well. On the other hand, CHARM-2 [66] produces both the bulk viscosity and slope of the curve in agreement with that reported by Brau and Jonkman [76] in their experimental study. However, their shear viscosity values show a clear deviation from that of NIST data. Out of all these models, the model by Javananien [67] is the only model that gives bulk viscosity matching with experiments in the complete range of 300 to 600 K. The deviation of shear viscosity predicted by this model is also small. Therefore, we have used this molecular model of oxygen gas for further study of the estimation of bulk viscosity of a mixture of nitrogen and oxygen gas.

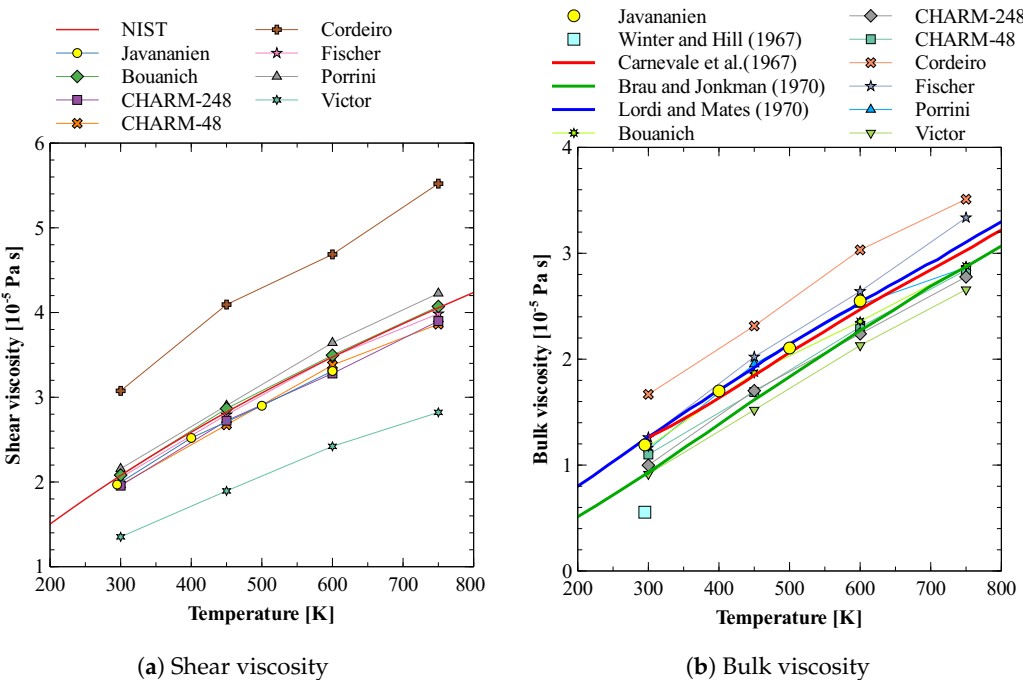

(**a**) Shear viscosity          (**b**) Bulk viscosity

**Figure 3.** Shear and bulk viscosity of $O_2$ calculated using different potentials [67–77].

### 5.4. Bulk Viscosity of $N_2 + O_2$ Mixture

Once we have optimum molecular models for both nitrogen and oxygen gases that produce both shear and bulk viscosities reasonably well, we use these models to study the mixture of nitrogen and oxygen gases. We calculate the bulk viscosity of this mixture at three different compositions, i.e., 25% $N_2$ + 75% $O_2$, 50% $N_2$ + 50% $O_2$, and 78% $N_2$ + 22 % $O_2$. The last composition is chosen such that it approximates the dry air. Figures 4a,b show the results obtained for shear and bulk viscosity of these mixtures [78]. It can be observed that the both the shear and bulk viscosity values vary linearly in the investigated temperature regime. Furthermore, for dry air, these estimates give bulk to shear viscosity ratio ($\mu_b/\mu$) between 0.6 to 0.8, as shown in Figure 5. Figure 5 also shows that the obtained $\mu_b/\mu$ ratio shows reasonably good agreement with previously reported values [23,79], especially those reported in the recent 2021 study by Ma et al. [23]. It should be noted that the experimental

bulk viscosity data given in Ref. [80] was calculated at pressure greater than 3 bar, whereas, in the present work simulations are performed at pressure 1 bar. The reported results on bulk viscosity are especially useful in simulations of high Mach number flow of air in earth atmosphere, where bulk viscosity effects cannot be neglected. Finally, it should also be noted that transport properties of mixtures are directly affected by the intermolecular interaction potential. Here, to obtain viscosity values, we have approximated the interaction between nitrogen and oxygen atoms using the Lorentz–Berthelot mixing rule. For an accurate estimation of viscosity values, potentials obtained through ab-initio calculations should be used.

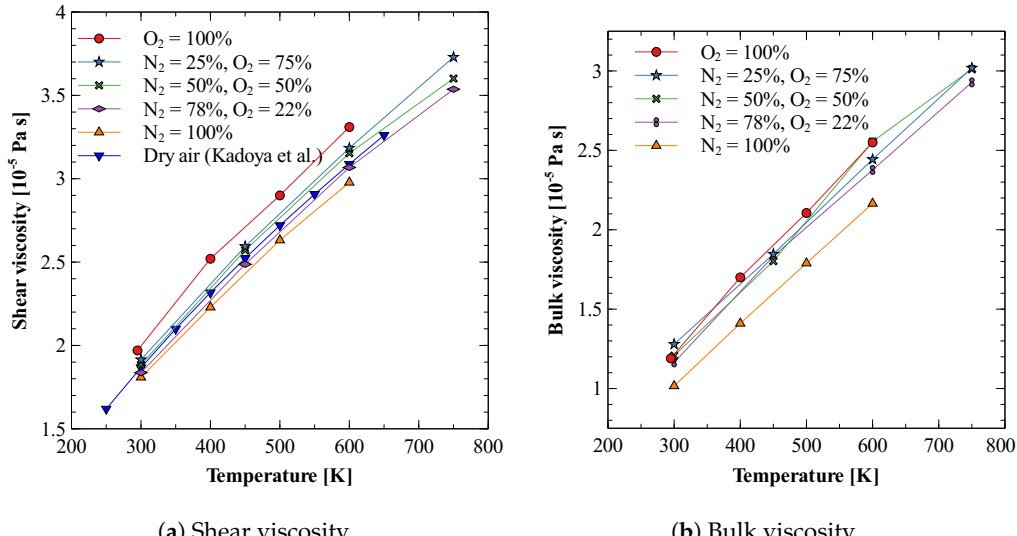

(**a**) Shear viscosity  (**b**) Bulk viscosity

**Figure 4.** Bulk and shear viscosity of $N_2$ and $O_2$ mixture at various compositions [78].

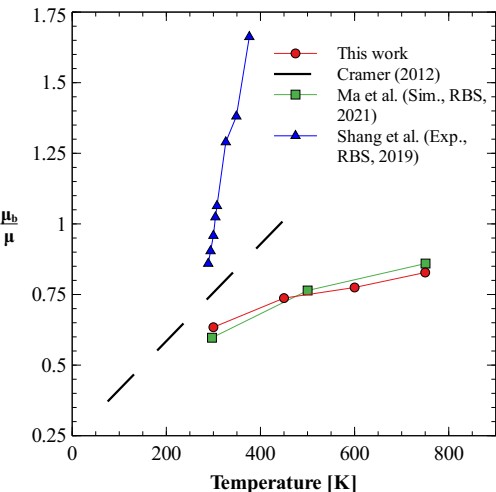

**Figure 5.** Bulk to shear viscosity ratio for dry air [23,79,80].

## 6. Conclusions

In this paper, we have studied the bulk viscosity of various gases and their mixtures, viz., isotopes of $N_2$; mixture of nitrogen with noble gases, i.e., $N_2$+Ne, $N_2$+Ar, $N_2$+Kr; pure $O_2$; mixture of $N_2$ and $O_2$; and mixture of $N_2$ + $H_2O$ gas mixtures. In the study of isotopes of nitrogen gas, it was observed that the bulk viscosity of gas increases with an increase in molecular mass, if the other molecule parameters, i.e., bond length and interaction parameters, are kept the same. However, bulk viscosity decreases if the total mass is kept constant but distributed nonuniformly among two constituent atoms of the diatomic nitrogen molecule. In the study of bulk viscosity of mixtures of nitrogen with

noble gases, it was observed that a small amount (5%) of impurity of noble gas does not alter bulk viscosity significantly, and bulk viscosity of 95% $N_2$ + 5% noble-gas mixture remained almost same as pure nitrogen. In contrast, a small amount (5%) of nitrogen in 95% noble gas makes a noticeable increase in the bulk viscosity of the mixture compared to the zero bulk viscosity of pure noble gases. Next, we reported the bulk viscosity of pure oxygen and mixtures of nitrogen and oxygen. Both nitrogen and oxygen were found to have almost similar values of bulk viscosity. Three different composition were considered, viz., 25% $N_2$ + 75% $O_2$, 50% $N_2$ + 50% $O_2$, and 78% $N_2$ + 22 % $O_2$.

Future works can aim to study the effect of humidity on bulk viscosity of nitrogen gas. This would require accurate modeling of vibrational levels accurately. However, it is not straightforward to simulate vibrational levels accurately in the classical molecular dynamics framework, as vibrational energy levels are quantized. Most of the potentials, such as Morse and harmonic potentials, treat vibrational levels in a continuous fashion. The present study is performed using classical molecular dynamics, and therefore, is limited to the calculation of bulk viscosity due to rotational relaxation only. The obtained bulk viscosity values can be directly applied to CFD simulations of moderate-temperature fluid flows, wherein the vibrational energy modes can be expected to play a negligible role.

**Author Contributions:** Conceptualization, B.S. and R.K.; methodology, B.S. and R.K.; software, B.S. and S.P.; validation, B.S., S.P. and R.K.; formal analysis, B.S., S.P. and R.K.; investigation, B.S. and S.P.; resources, S.P. and R.K.; data curation, B.S., S.P. and R.K.; writing—original draft preparation, B.S. and R.K.; writing—review and editing, B.S., S.P. and R.K.; visualization, B.S. and S.P.; supervision, R.K.; project administration, R.K. All authors have read and agreed to the published version of the manuscript.

**Funding:** This research received no external funding.

**Data Availability Statement:** The data presented in this study are available on request from the corresponding author.

**Acknowledgments:** The first two authors, B.S. and R.K., gratefully acknowledge the use of computational resources provided by DELL HPC and AI Innovation Lab for carrying out computations required for this work. The authors also acknowledge the National Supercomputing Mission (NSM) for providing computing resources of 'PARAM Sanganak' at IIT Kanpur, which is implemented by C-DAC and supported by the Ministry of Electronics and Information Technology (MeitY) and Department of Science and Technology (DST), Government of India.

**Conflicts of Interest:** The authors declare no conflict of interest.

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
