# Peer review of "Bulk Viscosity of Dilute Gases and Their Mixtures"

_fluids, doi:10.3390/fluids8010028_

Round 1
Reviewer 1 Report
The bulk viscosity of various dilute gases and their mixtures (nitrogen, oxygen, mixtures of nitrogen and noble gases, mixtures of nitrogen and oxygen) were calculated using the Green-Kubo method. The mechanism of bulk viscosity was shortly presented considering apparent and intrinsic bulk viscosity. Some applications of bulk viscosity were discussed in analytical and CFD models of various flow scenarios (20 publications were cited). The parameters of the molecular models for considered gases are given according to the data provided in the cited publications.
The results of the calculations were presented on graphs showing changes in viscosity (bulk and shear) as a function of temperature. Calculation results include: bulk and shear viscosity of different isotopes of nitrogen gas, bulk viscosity of different mixtures of nitrogen and noble gases (Ne, Kr, Ar), shear and bulk viscosity of oxygen (calculated using different potentials; the most suitable model was selected). Then the bulk and shear viscosity of N2 and O2 mixture at various compositions were calculated.
The manuscript is clear, relevant for the field and presented in a well-structured manner. The figures are appropriate and easy to interpret and understand. The cited references are current and relevant. The conclusions are consistent with arguments presented.
Substantive comments on the paper:
The calculation results of bulk and shear viscosity for different isotopes of nitrogen gas and for nitrogen and noble gases mixtures, have not been compared with any experimental data. This needs to be supplemented.
Also bulk and shear viscosity of N2 and O2 mixtures should be compared with experimental results. Reference was made to the work [23], but the quoted data (presented in Fig.5) come from a simulation, not from experiments. This needs to be supplemented.
The statement :
“Once we have optimized the Green-Kubo method for estimation of bulk viscosity…” (subsection 5.4. p.13)
is not confirmed in the work.
Methods of optimizing the adopted model are not described, it is not known what was optimized. Only the method of selecting a molecular model for oxygen was considered. This issue should be better explained and described.
Author Response
Please find attached the response to Reviewer 1. Thank you.

Reviewer 2 Report
My suggestion is given as follows: 1) The author should give the calculate process for the ploted results in the Figure 1a and 1b.2) Which group of parameters shown in Tables 1~2 are used, respectively?
Author Response
Please find attached the response to Reviewer 2. Thank you.
